# Selection and Evaluation of Reference Genes for RT-qPCR Analysis in *Amorphophallus* Konjac Based on Transcriptome Data

**DOI:** 10.3390/genes14081513

**Published:** 2023-07-25

**Authors:** Yanli Liu, Chengcheng Zhang, Nunung Harijati, Ying Diao, Erxi Liu, Zhongli Hu

**Affiliations:** 1College of Life Science and Technology, Wuhan Polytechnic University, Wuhan 430023, China; liuyanli3773@126.com (Y.L.); yingdiao@whpu.edu.cn (Y.D.); 2Lotus Engineering Research Center of Hubei Province, College of Life Science, Wuhan University, Wuhan 430072, China; ddjs221@163.com; 3Department of Biology, Faculty of Mathematics and Natural Sciences, Brawijaya University, Jl. Veteran, Malang 65145, Indonesia; nharijati@gmail.com; 4Institute of Konjac, Enshi Academy of Agricultural Sciences, Enshi 445000, China; liuerxi1985@163.com

**Keywords:** *Amorphophallus*, reference genes, RT-qPCR, gene expression

## Abstract

Combined with the Konjac transcriptome database of our laboratory and internal reference genes commonly used in plants, the eight candidate internal reference genes were screened and detected. They are the 25S ribosomal RNA gene (*25S rRNA*), 18S ribosomal RNA gene (*18S rRNA*), actin gene (*ACT*), glyceraldehyde-3-phosphate dehydrogenase gene (*GAPDH*), ubiquitin gene (*UBQ*), β-tubulin gene (*β-TUB*), eukaryotic elongation factor 1-αgene(*eEF-1α*), and eukaryotic translation initiation factor 4α-1 gene (*eIF-4α*). The results of GeNorm, Normfinder, and BestKeeper were analyzed comprehensively. The data showed that the expression levels of *25S rRNA*, *18S rRNA*, and *ACT* at the reproductive periods, *eEF-1α* and *eIF-4α* at the nutritional periods, and *eEF-1α*, *UBQ*, and *ACT* at different leaf developmental periods were stable. These identified and stable internal reference genes will provide the basis for the subsequent molecular biology-related studies of Konjac.

## 1. Introduction

*Amorphophallus* Blume ex Decne (*Araceae*) is primarily found in tropical or subtropical areas of South Asia and West Africa [1,2,3]. So far, more than 170 species have been found in this genus; some of them are used as medicine or food. *Amorphophallus* konjac is the only plant rich in Konjac glucomannan (KGM), the most representative water-soluble dietary fiber, which has a good effect on human health by reducing plasma lipids, blood pressure, glycemia, and even weight [4]. Identifying critical genes in the synthesis of Konjac KGM also provides valuable candidate genes for the molecular breeding of this crop [5]. Because *Amorphophallus* konjac is rich in KGM, it is widely cultivated in the central and western regions of China. Its cultivation area has already reached 26,000 hm^2^ [6]. The studies on Konjac mainly focus on the bioactive components extraction [7], molecular cloning [8], taxonomic and evolution [7,9], whole chloroplast genome sequencing [10], genetic molecular makers development [11], and comparative transcriptomics [2]. With the development of molecular biology research in Konjac, gene expression patterns had become an important part of gene functional study.

Real-time fluorescent quantitative PCR (RT-qPCR) has a series of advantages, comprising higher sensitivity, particularity, and wide quantification range, a common method for quantifying gene expression [12,13]. For quantifying quantitative amounts of gene expression, relative and absolute quantification techniques are frequently used. Nevertheless, one defect of relative quantification is that, at a minimum, one internal reference gene with steady expression was used as a control during the experiment. Internal reference genes, which are usually screened from housekeeping genes, were involved in cell formation and maintaining basic cellular functions such as protein folding and ribosome subunit synthesis. When appropriate reference genes were used as controls, the genes could be stably expressed under different experimental conditions [14]. The *β-TUB*, *25s rRNA*, *eEF-1α*, α-tubulin gene (*α-TUB*), *ACT*, *eIF-4α*, *GAPDH*, have been identified as dependable reference genes in plants [13,15]. Relative quantification was calculated based on the PCR data of target genes of stable reference genes [16]. Many studies have shown that sample initial RNA quality, reverse transcription efficiency, the specificity of primers, and the experimental environment could affect gene expression [17]. Such diverse abiotic stress conditions or various tissue development phases may have an impact on the expression level of reference genes. Therefore, in order to make the analysis results of RT-qPCR more accurate and reliable, it is very important to screen suitable internal reference genes for data correction and standardization. In recent years, many reference genes for RT-qPCR analysis have been selected in crops, such as *Miscanthus lutarioriparia* [18], switchgrass [19], wheat [20], tea [21], and cotton [22]. In the meantime, in two cultivated *Amorphophallus* species, the appropriate reference genes have been found in various tissues under stress from waterlogging or high temperatures [23]. The same material under different growth conditions and different experimental conditions may cause significant changes in gene expression. Therefore, the selection of stably expressed internal reference genes is very important for the accuracy of experimental results. However, there have been no studies on the internal reference genes of Konjac at different growth stages. In this study, combining the Konjac transcriptome database and common reference genes in plants, eight candidate reference genes were screened to analyze the steady gene expression in different tissues and different developmental stages of Konjac, hoping to provide a reference for subsequent molecular biology studies of Konjac.

## 2. Materials and Methods

### 2.1. Plant Sample Collection

The tested individuals of *A.* konjac were planted in a field with the same growth condition at Wuhan University, Hubei Province, China. Samples were taken at different periods. The plant materials included (1) eight tissues at the reproductive periods, (2) four tissues at the nutritional periods, (3) and leaves in three development periods (Table 1). Samples were taken separately for each of the 14 materials. Three biological replicates were taken for each part of each material, and three technical replicates were taken for determination. After sampling, each sample was washed with sterile water, and the surface water was dried with absorbent paper, and then immediately stored in liquid nitrogen in an ultra-low temperature refrigerator at −80 °C.

### 2.2. RNA Extraction and cDNA Synthesis

Total RNAs were extracted from dissimilar tissues at diverse periods using OminiPlant RNA Kit with DNase I (CW Biotech Co., Ltd., Beijing, China). The purity and also quality of RNA were determined by 1.2% agarose gel electrophoresis. The RNA quality is qualified if the bands are single, bright, clear, and have no trailing dispersion. The cDNAs (20 μL) were synthesized using the Fast Quant RT Kit with gDNase (Tiangen Biotech Co., Ltd., Beijing, China). The reverse transcription system was 20 μL, with 6 μL total RNA as the starting material, 10× RT Mix 2 μL, Super pure dNTPs 2 μL, Oligo-(dT) 2 μL, Quant reverse Transcriptase 1 μL, RNA-free water supplement 20 μL. Reverse recording: 37 °C for 15 min, 85 °C for 5 s, −20 °C storage.

### 2.3. Screening of Candidate Reference Genes, Primer Design, and RT-qPCR Assay

We screened eight candidate reference genes that are commonly used in other plants. The partial sequences of eight candidate reference genes were screened from available transcriptome data of Konjac in our laboratory to BLAST against the database in NCBI GeneBank to find the corresponding homologous single gene. Those unigenes of high identity score were blasted by ClustalW, then uses Primer Premier 5.0 software for primer design (Table 2). The required length was between 80 bp and 160 bp. Moreover, the selected candidate genes were cloned from Konjac and sequenced to verify the accuracy of transcriptome data.

The RT-qPCR assay was executed using the StepOne Real-Time PCR System (Applied Biosystem, Foster City, CA, USA). Every reaction (final volume of 20 μL) included 4.8 μL RNase-free ddH_2_O, 2 μL 50× ROX Reference Dye, 10 μL 2× SuperReal PreMix Plus, 2 μL diluted cDNA template, 0.6 μL of each primer. Two-step amplification included: pre-denaturation at 95 °C for 10 min, followed by 40 cycles of denaturation at 95 °C for 10 s, and annealing extension at 60 °C for 1 min. The melting curve analysis of each sample amplification product was used to verify the specificity of the PCR reaction. The cDNA of each sample was mixed in equal amounts, and the mixed stock solution was then diluted by gradient dilution (diluents 10^0^, 10^1^, 10^2^, 10^3^, 10^4^) for RT-qPCR reaction. Excel software was used to construct the standard curve, and the corresponding linear correlation coefficient R^2^ and the amplification efficiency (E) were obtained. RT-qPCR reaction using cDNA from 14 samples was performed to determine primer efficiency. Three technical and biological replicates were performed for all samples.

### 2.4. Data Analysis of Gene Expression Stability

The real-time quantitative PCR was performed using the StepOne Real-Time PCR System (Applied Biosystem, Foster City, CA, USA). The baseline threshold (StepOne Software v2.3) was set up manually to ensure the consistency of each sample with the matching tested gene placed in dissimilar reaction plates. The average, three technical replicates of Ct values with the parameter STD of <0.2 were calculated. The three programs geNorm, NormFinder, and BestKeeper, were used for the ranking comparison. The Ct values of three technical replicates were converted to relative expression quantities based on the formula of: 2^−ΔCt^ (ΔCt = the matching Ct value − minimum Ct value). In order to compare the comprehensive expression stability of the 8 candidate reference genes, geNorm, Normfinder, and BestKeeper programs were used to analyze and rank relative expression quantities of the 8 candidate reference genes, respectively.

## 3. Results

### 3.1. Primer Specificity and Amplification Efficiency of PCR Reaction

The sequencing results showed that the sequence of eight candidate reference genes obtained by cloning was consistent with those of the transcripts, suggesting the primers designed by the transcriptome data were reliable. The melting curves of the PCR amplification products were analyzed to verify the rationality of each pair of primers for the eight candidate reference genes (Figure 1). Melting curves showed a sole peak for each primer set indicating no non-specific amplification and the primer dimer. Amplification efficiency (E) among the test primers ranged from 90.19 to 103.73% with the correlation coefficient (R^2^) between 0.995 to 0.999 (Table 2 and Figure 2).

### 3.2. Ct Value Distribution and Expression Profile of the Eight Reference Genes

In this study, the precision of RT-qPCR for reference gene selection was assessed. The cDNA extracted from 14 tissues was mixed in equal amounts and served as the template for RT-qPCR to create a standard curve and measure the efficiency E of each internal gene amplification. The amplification efficiency of these 8 candidate reference genes ranged from 90% to 110%, and all curves showed a unimodal fusion peak, which proved that the selected candidate internal reference genes had good specificity, met the standard of RT-qPCR, and could be used for the next test. The relationship between the Ct value and the gene’s level of expression was inverse. The relative expression levels of the eight candidate internal reference genes were calculated using Ct values. According to earlier research, appropriate reference genes should be highly expressed and display steady expression levels across multiple samples. The selected genes showed Ct values ranging from 13.87 to 32.08. Among the tested genes, *25S rRNA*, and *18S rRNA* showed both the highest expression level and the lowest transcript abundance. The trend of expression level for the rest genes (*ACT*, *GAPDH*, *UBQ*, *βTUB*, *eEF-1α*, *eIF-4α*) was similar in different Konjac tissues (Figure 3).

Cycle threshold (Ct) values of the calculated reference genes of all samples used in the experiments. Raw Ct values of fourteen samples were shown by a box plot. The box plot graph of the Ct value shows the mean values as a line across the box. Short-term represented the maximum and minimum values. Lower and upper boxes indicated the 25th percentile to the 75th percentile.

### 3.3. Stability Analysis of Reference Genes by GeNorm

GeNorm was used to detect the two or more most suitable reference genes. The M value of candidate reference genes was obtained. The M value is oppositely proportional to gene stability. Generally, M = 1.5 was the critical value. If M < 1.5, it was considered to have better expression stability and was suitable as a reference gene. The consequences of geNorm analysis were exhibited in Table 3. Both *25S rRNA* and *18S rRNA* were with the lowest M value of 0.513, suggesting they might be the most stable genes in the expression profile at the reproductive phase. The stability of the rest genes was *eEF-1α > UBQ > eIF-4α > ACT > GAPDH > β-TUB* (Figure 4A).

The pairwise variations (Vn/n + 1) were also calculated with geNorm between two sequential ranked genes to determine the minimum number of reference genes. If Vn/n + 1 < 0.15, the suitable number of the finest reference genes should be n; if the Vn/n + 1 > 0.15 or Vn/n + 1 = 0.15, the number should be n + 1. At the reproductive phases, it can be seen in Figure 4A that the V value of V3/V4 was 0.149 < 0.15. The three most stable internal reference genes were selected at the reproductive phases, including *25S rRNA*, *18S rRNA*, and *ACT*, to make the experimental data more reliable. According to the above method, *eEF-1α* and *eIF-4α* with the lowest M value of 0.310 were the most stable reference genes at the nutritional phases, while *25S rRNA* with an M value of 1.097 was the minimum stable reference gene under the same phase (Figure 4B). It as well as can be seen in Figure 4B that the V value of V2/V3 was 0.135 < 0.15. At the vegetative phases, the two most steady internal reference genes were selected, including *eEF-1α* and *eIF-4α* to make the experimental data more reliable. In addition, the steadiness of the eight internal reference genes was in descending order from high to low: *eEF-1α* = *UBQ* (M = 0.291) > *ACT* > *β-TUB* > *GAPDH* > *18S rRNA* > *eIF-4α* > *25S rRNA* (M = 0.935) (Figure 4C). Furthermore, it can be seen in Figure 4C that the V value of V2/V3 was 0.135 < 0.15. In the three developmental stages of the leaf, the two most steadiness internal reference genes were selected, including *eEF-1α* and *eIF-4α*, to make the experimental data more reliable.

### 3.4. Stability Analysis of Reference Genes by Normfinder

The Normfinder software assesses the stability of each candidate reference gene based on specific experimental conditions. Normfinder software can effectively solve the problem that geNorm software can not statistically distinguish the use of genes with similar expression patterns. Normfinder can compare not only the differences between candidate genes but also the intergroup variation in the sample. It is based on a mathematical model for individual analysis of sample subgroups and computations of intra-group expression changes based on stability values (SV). The lower the stability value, the more stable the candidate internal reference gene.

The results of the Normfinder analysis showed in Table 3. The SV of *25S rRNA*, *18S rRNA*, *ACT*, *GAPDH*, *UBQ*, *β-TUB*, *eEF-1α*, and *eIF-4α* from eight different tissues at the reproductive phase was 0.354, 0.336, 0.448, 0.630, 0.375, 0.745, 0.075, and 0.532, respectively (Figure 5A). From this result, the *eEF-1α* (SV = 0.075) had the highest stability, followed by *18S rRNA* (SV = 0.336) and *25S rRNA* (SV = 0.354). Thus, *eEF-1α*, the most stable reference gene, was advised to use in the reproductive phases. Similar to the above method, *25S rRNA* (SV = 0.987) was ranked as the least stable gene, while the *eEF-1α* (SV = 0.274) was a stable gene, followed by *UBQ* (SV = 0.344) and *18S rRNA* (SV = 0.375) (Figure 5B). So, the *eEF-1α* was the most suitable one at the nutritional phases. Figure 5C showed that *25S rRNA* (SV = 0.834) was the least stable gene, while *β-TUB* (SV = 0.161) was the most stable, followed by *ACT* (SV = 0.260) and *eEF-1α* (SV = 0.305). The NormFinder results indicated that *β-TUB* was the most suitable reference gene at three developmental stages of the leaf.

### 3.5. Stability Analysis of Reference Genes by BestKeeper

BestKeeper software was used to analyze the expression levels of internal reference genes in different tissues. The software could directly use Ct values for analysis without converting Ct values into relative expressions. The correlation coefficient r, coefficient of variation CV, and standard deviation SD were obtained by the software. The larger the r, the smaller the CV and SD, and the more stable the internal reference gene expression.

The results of the BestKeeper analysis were shown in Table 3. The results showed that the SD values of *ACT*, *GAPDH*, and *β-TUB* > 1 with low stability were unsuitable as reference genes at the reproductive phase. The correlation coefficient r value of *25S rRNA* and *18S rRNA* was relatively larger with a small SD value and CV, indicating that these two genes were suitable for synergistic use as reference genes. Comprehensive analysis showed that *25S rRNA* and *18S rRNA* were suitable for use as reference genes during the reproductive phases. At the nutritional phases, *eEF-1α* has a large correlation coefficient R and a minimum SD value and CV, indicating that gene expression stability between samples was strong and recommending that *eEF-1α* was suitable for the reference gene. In the end, *eIF-4α* has a minimum correlation coefficient r (−0.11) and a minimum SD value (0.44) and CV (1.57), suggesting that the expression of *eIF-4α* was stable between samples and suitable for reference alone at the three developmental stages of the leaf.

## 4. Discussion

Compared with traditional PCR, RT-qPCR has the advantages of high specificity, rapidity, and sensitivity, and was often used as an effective tool to detect gene expression. The selection of appropriate internal reference genes can make RT-qPCR data of target genes more reliable, but inappropriate internal reference genes will lead to the deviation of experimental analysis results. Therefore, in order to ensure the accuracy of test results, it is necessary to introduce stable expression of internal reference genes. Due to housekeeping genes participate not only in primary cell metabolism but also in other cellular functions, even the same housekeeping gene might show different stability in gene expression among different species [18]. In different samples, different tissues from the same sample, or even different locations from the same tissue, the expression of housekeeping genes varied greatly [24,25,26,27]. On the one hand, gene expression shows temporal specificity, which means the expression of a gene rigorously follows a particular time sequence, leading to changes in gene expression during different growth stages. On the other hand, it also has spatial specificity, which was established by the differential cell distribution in various organs. Gene expression can be regulated to sustain cell division and differentiation, individual growth and development, and a better ability for organisms to adapt to changes in their surroundings. So, the housekeeping genes are not constant for gene expression study and must be chosen depending on the sample and for verification purposes, because its expression is influenced by the promoter sequence or the interaction between the promoter and RNA polymerase. The stable expression of internal reference genes is the basic prerequisite for standardizing the expression profile of target genes. Real-time fluorescence quantitative PCR is an effective method to analyze gene expression. However, the analysis results are influenced by many factors, such as RNA quality, RNA reverse transcription, and amplification efficiency. Although stable reference genes have been established in many crops, there is little research on the internal reference gene screening of *Amorphophallus* [23].

Some genes can still be stably expressed even if the environment changes greatly, while the expression of other genes will change dramatically [28]. This further demonstrates the importance of selecting stably expressed internal reference genes in different tissues or under different treatment conditions. There are numerous bio-pathways and genes involved in the Konjac life span waiting for our exploration. Therefore, screening the best internal reference genes is important to molecular genetic research in Konjac. It was the first time to investigate the most reliable reference genes for RT-qPCR analysis in Konjac at the nutritional, reproduction, and different leaf developmental phases. The comprehensive results demonstrated that the suitable reference genes might vary depending on the species or other unique experimental circumstances.

## 5. Conclusions

To date, this research is the first work that aims to use RT-qPCR to verify suitable candidate reference genes in the reproduction phase, vegetative phase, and different development stages of Konjac leaves to realize the standardization of gene expression. In conclusion, the suitable internal reference genes were *25S rRNA*, *18S rRNA*, and *ACT* at the reproductive phase, *eEF-1α* and *eIF-4α* at the nutritional stages, and *eEF-1α*, *UBQ*, and *ACT* at different leaf development stages. These consequences may support a beginning point to select reference genes henceforth.

## Figures and Tables

**Figure 1 genes-14-01513-f001:**
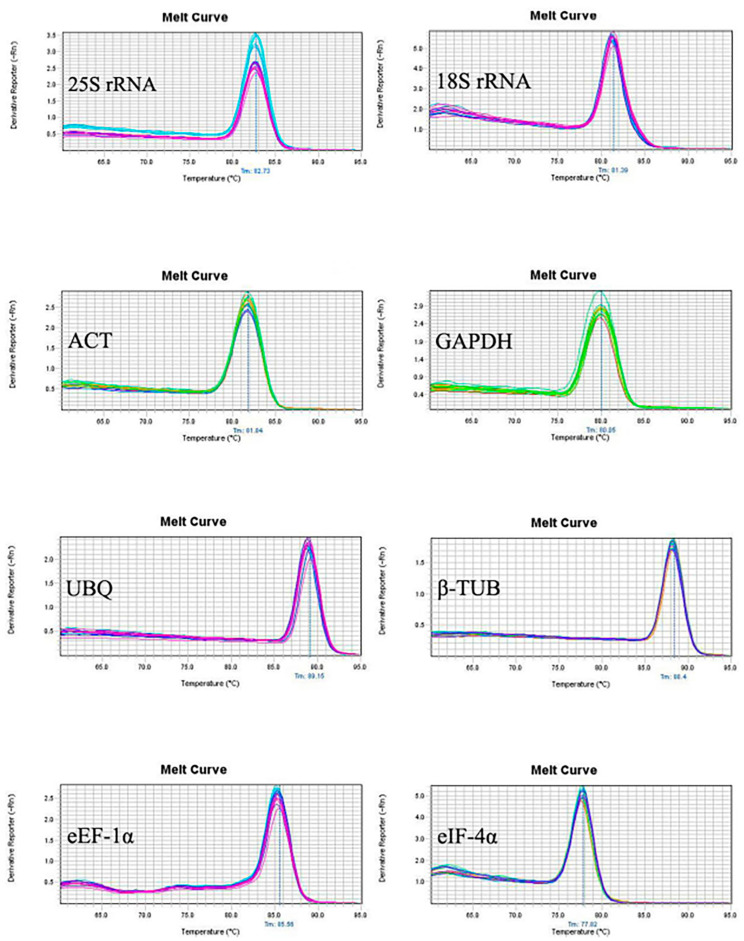
The melting curves corresponding to the 8 pairs of primers were only single-peak, primer-free dimers, and non-specific bands were generated, which proved that the selected candidate reference genes had good specificity and met the RT-qPCR standard.

**Figure 2 genes-14-01513-f002:**
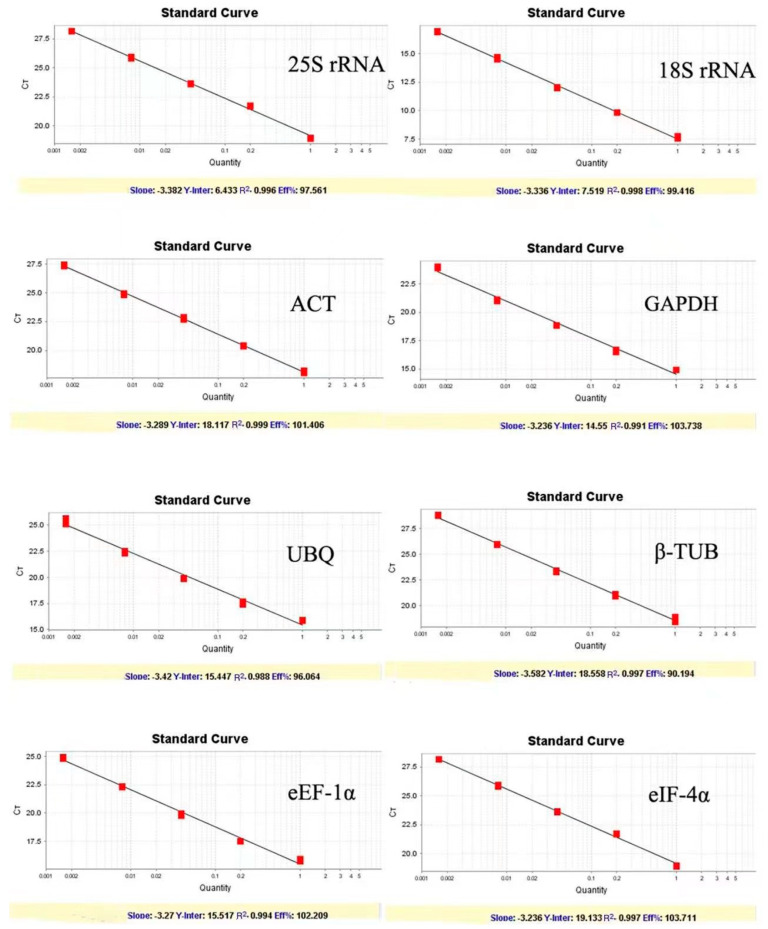
Standard curves of the 8 candidate housekeeping genes.

**Figure 3 genes-14-01513-f003:**
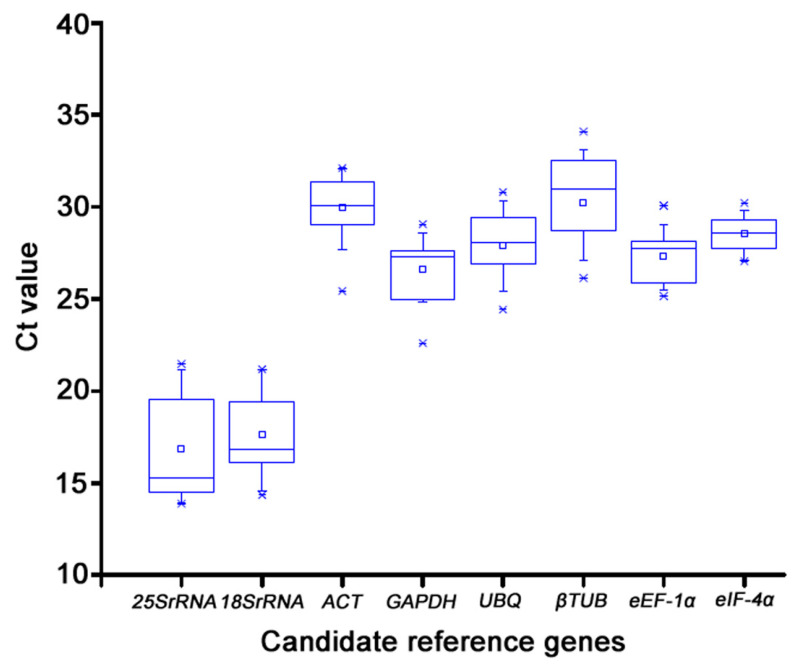
The CT values of 8 candidate reference genes.

**Figure 4 genes-14-01513-f004:**
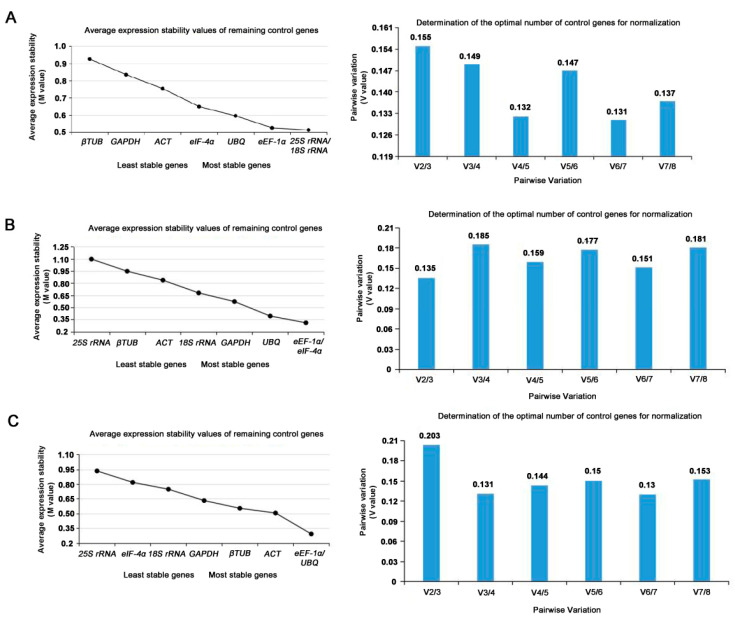
The gene expression steadiness values (M) (**Left**) and the pairwise variation value (V) (**Right**) of the candidate reference gene calculated by geNorm. (**A**). Sequence the stability of candidate reference genes that are processed and expressed during the breeding period. The least stable genes are on the left and the most stable genes are on the right. The paired variation (Vn/Vn + 1) between the normalized factor NFn and NFn + 1 at the reproductive stage was analyzed. (**B**). Sequence the stability of candidate reference genes that are treated and expressed during the vegetative phase. The most unstable genes are on the left and the most stable genes are on the right. The paired variation (Vn/Vn + 1) between the normalized factor NFn and NFn + 1 was analyzed. (**C**) Sequence the stability of candidate reference genes that were processed and expressed at different stages of leaf development. The most unstable genes are on the left and the most stable genes are on the right. Pairwise variation (Vn/Vn + 1) between standardized factors NFn and NFn + 1 at different developmental stages of leaves was analyzed.

**Figure 5 genes-14-01513-f005:**
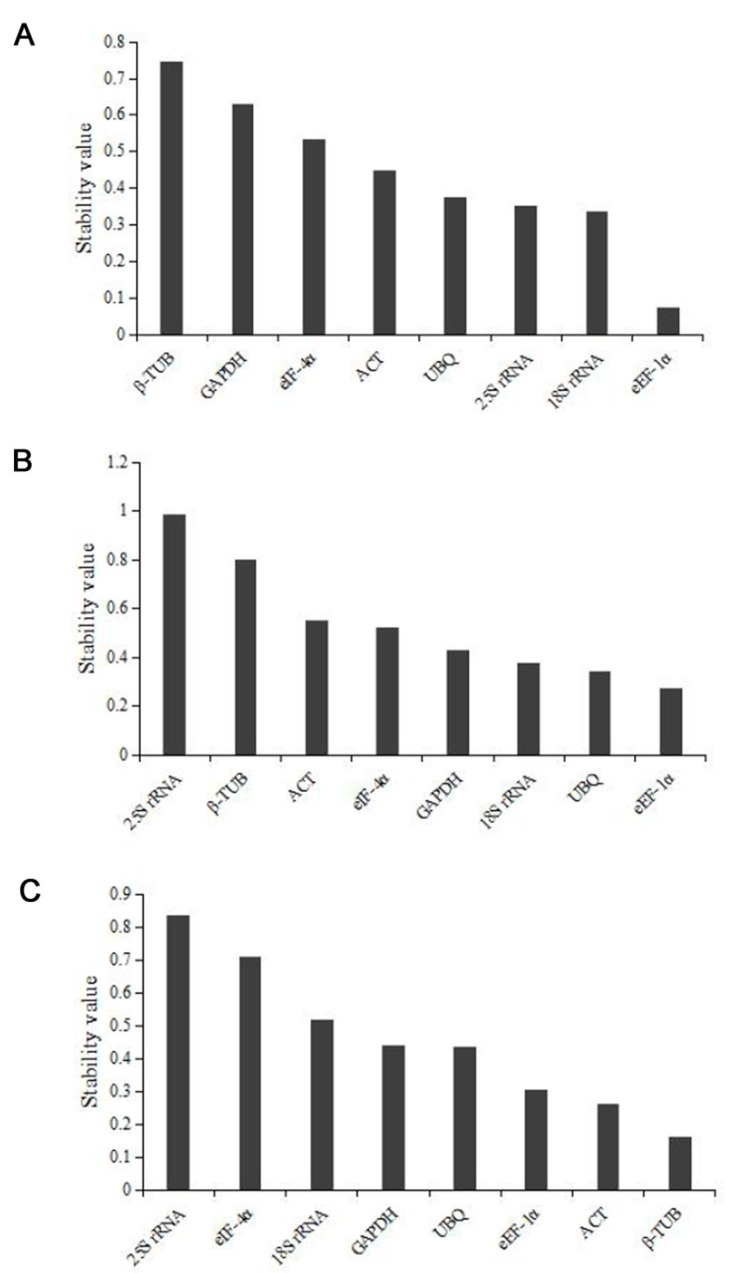
Average expression stability values (S) calculated by Normfinder at (**A**) reproductive phase, (**B**) vegetative phase, and (**C**) different developmental stages of leaves.

**Table 1 genes-14-01513-t001:** Plant materials used in this study.

Growing Stage	Spathe	Append Age	Stamen	Pistil	Inflorescence Axis	Inflorescence Stalk	Tuber	Root	Leaf	Petiole
Reproductive period	√	√	√	√	√	√	√	√		
Nutritional growing period							√	√	√	√
Germinating period									√	
Leaf opening period									√	
Leaf mature period									√	

**Table 2 genes-14-01513-t002:** Description of candidate reference genes and their primer sequences used for RT-qPCR analysis.

Gene Name	Sequence Primer 5′-3′	Size (bp)	PCR Efficiency (%)	R^2^
*25S rRNA*	F: CGCCCTCGACCTATTCTCAAA R: CTTACCAAAAATGGCCCACTT	103	97.56	0.995
*18S rRNA*	F: AGACGAACAACTGCGAAAGC R: GGCGGAGTCCTAAGAGCAAC	149	99.41	0.998
*ACT*	F: TGAACGTGAAATTGTAAGGGAC R: CAGATGAGCTAGTCTTGGCAGT	95	101.40	0.999
*GAPDH*	F: AGAGGAGCGAGGCAGTTAGTG R: CCCATGTTTGTTGTTGGTGTA	95	103.73	0.991
*UBQ*	F: CCAGCAGCGCCTCATCTTTG R: CTTGGGCTTGGTGTAGGTCTTC	151	96.06	0.998
*β-TUB*	F: CGGATGATGCTGACCTTCTCG R: ATGCACTCGTCGGCGTTCTC	116	90.19	0.997
*eEF-1α*	F: CTGAAGAATGGCGATGCTGG R: ACCGTCTGCCTCATGTCCCT	119	102.20	0.994
*eIF-4α*	F: AGCATTTCATCCGCTTCGTC R: TTGTGGGTACTCCTGGTCGT	100	103.71	0.997

R^2^ the correlation coefficients of the standard curve.

**Table 3 genes-14-01513-t003:** Gene expression stability ranked by geNorm, Normfinder, and BestKeeper.

Different Developmental Stage of Konjac	Gene Name	GeNorm Stability (M)	Normfinder Stability (SV)	BestKeeper		Stability
				**SD**	**CV%**	**r**
Reproductive phase	*25S rRNA*	0.513	0.354	0.56	3.83	0.89
	*18S rRNA*	0.513	0.336	0.70	4.38	0.91
	*ACT*	0.754	0.448	1.64	5.58	0.95
	*GAPDH*	0.835	0.630	1.64	6.27	0.93
	*UBQ*	0.595	0.375	0.90	3.4	0.91
	*β-TUB*	0.926	0.745	1.17	4.11	0.81
	*eEF-1α*	0.525	0.075	0.81	3.06	0.86
	*eIF-4α*	0.649	0.532	0.69	2.38	0.83
Nutritional phase	*25S rRNA*	1.097	0.987	1.27	6.51	0.99
	*18S rRNA*	0.679	0.375	1.08	5.37	0.98
	*ACT*	0.836	0.552	0.61	2.01	0.50
	*GAPDH*	0.572	0.430	0.90	3.38	0.82
	*UBQ*	0.393	0.344	0.42	1.41	0.50
	*β-TUB*	0.947	0.802	0.75	2.34	0.28
	*eEF-1α*	0.310	0.274	0.24	0.86	0.88
	*eIF-4α*	0.310	0.521	0.31	1.11	−0.55
Three developmental stages of leaves	*25S rRNA*	0.935	0.834	1.11	5.62	0.82
	*18S rRNA*	0.748	0.517	0.46	2.35	0.85
	*ACT*	0.505	0.260	1.05	3.38	0.98
	*GAPDH*	0.632	0.440	0.83	2.97	1.00
	*UBQ*	0.291	0.434	0.55	1.82	0.60
	*β-TUB*	0.553	0.161	1.17	3.58	0.96
	*eEF-1α*	0.291	0.305	0.74	2.56	0.85
	*eIF-4α*	0.818	0.711	0.44	1.57	−0.11

## Data Availability

No new data were created.

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
