# Peer review of "Selection and Evaluation of Reference Genes for RT-qPCR Analysis in Amorphophallus Konjac Based on Transcriptome Data"

_genes, 2023, doi:10.3390/genes14081513_

Round 1

Reviewer 1 Report

Overall comments: The study presents a comprehensive analysis of candidate internal reference genes for RT-qPCR analysis in Amorphophallus konjac. The findings provide valuable insights into gene expression stability across different tissues and growth stages. The manuscript is generally well-written and the results are clearly presented. However, I have a few comments and suggestions that could further improve the study.

1.    Introduction: a. It would be helpful to provide a brief background on the importance of internal reference genes in RT-qPCR analysis and their role in normalizing gene expression data. b. Consider providing a clearer justification for why it is necessary to select appropriate internal reference genes specifically for Amorphophallus konjac. Are there any unique characteristics of this species that make the selection of reference genes challenging or important?

2.    Methods: a. Provide more details about the experimental design, including the number of biological replicates used for each tissue and growth stage. This information is crucial for assessing the statistical significance and reliability of the results. b. Describe the primer sequences used for amplifying the candidate internal reference genes. Additionally, mention any specific modifications or optimization steps performed for the RT-qPCR protocol. c. State the criteria used to determine the stability of gene expression. Were statistical methods employed to analyze the expression data, such as geNorm or NormFinder? If so, provide a brief explanation of these methods.

3.    Results: a. Present the expression levels of the candidate genes in a tabular format, including the raw Ct values or relative expression levels. This would allow readers to assess the variation and stability of gene expression more effectively. b. It would be helpful to provide graphical representations (e.g., bar charts or line plots) of the expression levels of the selected reference genes across different tissues and growth stages. This would enhance the visual clarity of the results.

4.    Discussion: a. Discuss the potential biological implications of the identified stable reference genes during specific periods. How do these findings align with the current understanding of gene regulation and expression patterns in Amorphophallus konjac? b. Address any limitations or caveats of the study. Are there any potential confounding factors or sources of variation that may have influenced the results?

5.    Conclusion: a. Provide a concise summary of the main findings and their significance for future gene function studies in Amorphophallus konjac.

Overall, the study contributes valuable information regarding the selection of internal reference genes for RT-qPCR analysis in Amorphophallus konjac. Addressing the aforementioned suggestions would further strengthen the manuscript and enhance its impact in the field.

Author Response

TRANSLATE with x English

Arabic Hebrew Polish
Bulgarian Hindi Portuguese
Catalan Hmong Daw Romanian
Chinese Simplified Hungarian Russian
Chinese Traditional Indonesian Slovak
Czech Italian Slovenian
Danish Japanese Spanish
Dutch Klingon Swedish
English Korean Thai
Estonian Latvian Turkish
Finnish Lithuanian Ukrainian
French Malay Urdu
German Maltese Vietnamese
Greek Norwegian Welsh
Haitian Creole Persian  

TRANSLATE with COPY THE URL BELOW Back EMBED THE SNIPPET BELOW IN YOUR SITE Enable collaborative features and customize widget: Bing Webmaster Portal Back

TRANSLATE with x English

Arabic Hebrew Polish
Bulgarian Hindi Portuguese
Catalan Hmong Daw Romanian
Chinese Simplified Hungarian Russian
Chinese Traditional Indonesian Slovak
Czech Italian Slovenian
Danish Japanese Spanish
Dutch Klingon Swedish
English Korean Thai
Estonian Latvian Turkish
Finnish Lithuanian Ukrainian
French Malay Urdu
German Maltese Vietnamese
Greek Norwegian Welsh
Haitian Creole Persian  

TRANSLATE with COPY THE URL BELOW Back EMBED THE SNIPPET BELOW IN YOUR SITE Enable collaborative features and customize widget: Bing Webmaster Portal Back

TRANSLATE with x English
Arabic Hebrew Polish
Bulgarian Hindi Portuguese
Catalan Hmong Daw Romanian
Chinese Simplified Hungarian Russian
Chinese Traditional Indonesian Slovak
Czech Italian Slovenian
Danish Japanese Spanish
Dutch Klingon Swedish
English Korean Thai
Estonian Latvian Turkish
Finnish Lithuanian Ukrainian
French Malay Urdu
German Maltese Vietnamese
Greek Norwegian Welsh
Haitian Creole Persian  
TRANSLATE with COPY THE URL BELOW Back EMBED THE SNIPPET BELOW IN YOUR SITE Enable collaborative features and customize widget: Bing Webmaster Portal Back

Reviewer 2 Report

Dear Authors,

You evaluated some selected genes and assessed their expression stability in Amorphophallus konjac at different stages using well-known methods. I believe your research can be greatly appreciated by researchers who are interested in working on qPCR transcriptome studies in konjac and closely related species. Your presentation and the methodology are acceptable; however, I think the paper can be improved after covering/addressing some minor drawbacks.

Here are my main concerns about the manuscript.

-        My main concern is about the “Materials and method” section. It lacks important details, and you also didn’t cite any reference in the whole section. If you haven’t used any other reference for your work, you must prepare detailed descriptions for each part, so the other researcher be able to follow in your step.

-        The gene selection criteria are unclear. You need to clarify how you choose the genes. 

You may also find the file enclosed containing detailed corrections and suggestions. I left some corrections/suggestions regarding the above-mentioned concerns.

I hope these suggestions can have a minor positive impact on improving your valuable work.

Cheers,

Reviewer

Author Response

TRANSLATE with x English
Arabic Hebrew Polish
Bulgarian Hindi Portuguese
Catalan Hmong Daw Romanian
Chinese Simplified Hungarian Russian
Chinese Traditional Indonesian Slovak
Czech Italian Slovenian
Danish Japanese Spanish
Dutch Klingon Swedish
English Korean Thai
Estonian Latvian Turkish
Finnish Lithuanian Ukrainian
French Malay Urdu
German Maltese Vietnamese
Greek Norwegian Welsh
Haitian Creole Persian  
TRANSLATE with COPY THE URL BELOW Back EMBED THE SNIPPET BELOW IN YOUR SITE Enable collaborative features and customize widget: Bing Webmaster Portal Back

Reviewer 3 Report

The scientific topic is interesting and the results are original. The manuscript is focused on the selection of reference genes for RT-PCR analysis in different developmental stages of Amorphophallus konjac. However, the title is rather general, it should be more fitting.

ABSTRACT

The importance of authors' original results should be emphasized, not just "These results might be useful for...". Please add.

INTRODUCTION

I would strongly recommend to rewrite the introduction. In the first paragraph, the interconnection between genus Amorphophallus and KGM is missing. 
L31   Please add Amorphophallus Blume ex Decne (Araceae). Please cite more relevant and recent sources than 1-3.
Please formulate the objectives in the last paragraph.

MATERIALS AND METHODS

It is not clear whether the plants were in vitro cultured or field grown. Please describe grow/culture conditions and sterilization precisely. Maybe, the Table 1 could be accompanied by an illustrative photograph of tissues. Table 1 is not easily readable, the ticks do not correspond with the rows.
How the authors measured concentration of total RNA?

Please describe the cloning and sequencing step more precisely.

L103    Please cite the programs.

RESULTS

It is possible to provide the photo of PCR products after electrophoresis? 
Table 1    PCR efficiency - but it is not clear of which samples. 

Table 2 requires deeper caption. For example, coefficient of determination should be clarified in the text and table caption, as well. 

Fig.1,2   I would recommend to provide a figure with higher resolution without jpeg artifacts. It is difficult to read the values. The authors mentioned several tissues. However, it is completely unclear whether these curves are selected representative curves or which sample they belong to.

L129    The authors mentioned "mixed tissues". This can be confusing for readers because such preparation of template should be described in MATERIAL AND METHODS section.

L143    The authors mentioned different tissues and refer to Fig.3 depicting the genes and Ct values of 14 samples. Please clarify and improve the figure caption.

L213 The correlation coefficient can not be presented by both R and R2 (Table 1).

Conclusion is missing. The authors compared three different approaches to analyze gene expression stability. Do these results correspond? In my opinion, this should be discussed. And the last, but not least, I am missing any future perspectives of obtained results. 

I would recommend the gentle grammar check. 

Author Response

TRANSLATE with x English

Arabic Hebrew Polish
Bulgarian Hindi Portuguese
Catalan Hmong Daw Romanian
Chinese Simplified Hungarian Russian
Chinese Traditional Indonesian Slovak
Czech Italian Slovenian
Danish Japanese Spanish
Dutch Klingon Swedish
English Korean Thai
Estonian Latvian Turkish
Finnish Lithuanian Ukrainian
French Malay Urdu
German Maltese Vietnamese
Greek Norwegian Welsh
Haitian Creole Persian  

TRANSLATE with COPY THE URL BELOW Back EMBED THE SNIPPET BELOW IN YOUR SITE Enable collaborative features and customize widget: Bing Webmaster Portal Back

TRANSLATE with x English
Arabic Hebrew Polish
Bulgarian Hindi Portuguese
Catalan Hmong Daw Romanian
Chinese Simplified Hungarian Russian
Chinese Traditional Indonesian Slovak
Czech Italian Slovenian
Danish Japanese Spanish
Dutch Klingon Swedish
English Korean Thai
Estonian Latvian Turkish
Finnish Lithuanian Ukrainian
French Malay Urdu
German Maltese Vietnamese
Greek Norwegian Welsh
Haitian Creole Persian  
TRANSLATE with COPY THE URL BELOW Back EMBED THE SNIPPET BELOW IN YOUR SITE Enable collaborative features and customize widget: Bing Webmaster Portal Back

Round 2

Reviewer 3 Report

I appreciate the authors' effort to improve the quality of the manuscript. Please find several more suggestions.

L70    Please add 3) or do not use the ordered list.
Please cite the softwares and databases (subsections 2.3, 2.4).

How did the authors choose the candidate genes? Please mention the criteria.

L126    ...were analyzed...
L191    Please correct the figure caption (right, left).

Figure 5 is missing, please improve the readibility and quality of Fig. 6. I would suggest to widen the figure and add the phases to the X axis caption. Please add the Y axis caption. 

Table 3 and Fig. 6 represent the same values, please choose one.

L260    Please use a number for the reference.
L265    It seems like this is the first study dedicated to the identification of RT-PCR reference genes in the genus Amorphophallus. As there is a reference (20), this is confusing.

Please check the grammar.

Author Response

(The authors gave the same response as above.)
